# Supplementation with Fish Oil and Selenium Protects Lipolytic and Thermogenic Depletion of Adipose in Cachectic Mice Treated with an EGFR Inhibitor

**DOI:** 10.3390/cells13171485

**Published:** 2024-09-04

**Authors:** Hang Wang, Yi-Lin Chan, Yi-Han Chiu, Tsung-Han Wu, Simon Hsia, Chang-Jer Wu

**Affiliations:** 1Department of Nutrition, Hung-Kuang University, Taichung 433304, Taiwan; 2Department of Life Science, Chinese Culture University, Taipei 111396, Taiwan; phd.elainechan@gmail.com; 3Department of Microbiology, Soochow University, Taipei 111002, Taiwan; chiuyiham@scu.edu.tw; 4Division of Hemato-Oncology, Department of Internal Medicine, Chang Gung Memorial Hospital, Keelung 204006, Taiwan; u402026@gmail.com; 5Department of Food Science and Center of Excellence for the Oceans, National Taiwan Ocean University, Keelung 202301, Taiwan; 6Taiwan Nutraceutical Association, Taipei 104483, Taiwan; dr.simon.hsia@gmail.com; 7Graduate Institute of Medicine, Kaohsiung Medical University, Kaohsiung 807378, Taiwan

**Keywords:** white adipose tissue browning, cachexia, fish oil, selenium yeast, target therapy

## Abstract

Lung cancer and cachexia are the leading causes of cancer-related deaths worldwide. Cachexia is manifested by weight loss and white adipose tissue (WAT) atrophy. Limited nutritional supplements are conducive to lung cancer patients, whereas the underlying mechanisms are poorly understood. In this study, we used a murine cancer cachexia model to investigate the effects of a nutritional formula (NuF) rich in fish oil and selenium yeast as an adjuvant to enhance the drug efficacy of an EGFR inhibitor (Tarceva). In contrast to the healthy control, tumor-bearing mice exhibited severe cachexia symptoms, including tissue wasting, hypoalbuminemia, and a lower food efficiency ratio. Experimentally, Tarceva reduced pEGFR and HIF-1α expression. NuF decreased the expression of pEGFR and HIF-2α, suggesting that Tarceva and NuF act differently in prohibiting tumor growth and subsequent metastasis. NuF blocked LLC tumor-induced PTHrP and expression of thermogenic factor UCP1 and lipolytic enzymes (ATGL and HSL) in WAT. NuF attenuated tumor progression, inhibited PTHrP-induced adipose tissue browning, and maintained adipose tissue integrity by modulating heat shock protein (HSP) 72. Added together, Tarceva in synergy with NuF favorably improves cancer cachexia as well as drug efficacy.

## 1. Introduction

Lung cancer is the leading cause of cancer-related mortality worldwide (1.4 million deaths annually) with a 5-year survival rate of approximately 14%. Non-small cell lung cancer (NSCLC) represents ~80% of lung cancer cases [1,2]. Chemotherapy and/or irradiation treatments usually fail because NSCLC cells are intrinsically resistant to them. Chemotherapy is quite ineffective for patients with advanced NSCLC with only a 20%–35% response rate and a 10- to 12-month median survival [3]. Erlotinib (Tarceva), an EGFR tyrosine kinase inhibitor with clinical effectiveness, was approved to treat several different cancers [2]. Tarceva was shown to be positive in treating advanced NSCLC, while it remains challenging when speaking of a satisfactory therapeutic effect.

Patients with advanced lung cancer are highly susceptible to weight loss and cachexia. An estimated 60% of patients with lung cancer has cancer cachexia [4]. Patients affected typically experience poor nutrition, systemic inflammation, body weight loss, adipose tissue atrophy, skeletal muscle wasting, low therapy tolerance, and increased susceptibility to infections. Cancer cachexia is a negative risk factor concerning cancer survival, causing frailty in patients and often preventing them from undergoing further therapies [5]. No effective treatment is currently available for cachexia, which is responsible for approximately 20% of total deaths of patients with cancer [6]. Therefore, new therapeutics for cachexia prevention and treatment are urgently needed.

Metabolic dysfunction and an increased metabolic rate have been proposed to cause cancer cachexia. The metabolic changes seen in cachexia are a result of higher level of resting energy expenditure. Most cachexia research is focused on muscle wasting, while the true fact is that the depletion of adipose tissue (AT) sabotages metabolism and overall energy balance [7]. The parathyroid hormone-related protein (PTHrP) and interleukin-6 (IL-6), which act as inducers, provoke cancer cachexia [8,9]. Kir et al. demonstrated that tumor-derived PTHrP-stimulated expression of thermogenic genes in AT and increased resting energy expenditure in mice by browning white ATs (WATs) [8]. Decreasing WAT browning by silencing tumor production of IL-6 ameliorated severity of cachexia [9], highlighting a new role of PTHrP and IL-6 in hypermetabolism mediated malignancy. Moreover, the expression of heat shock proteins (HSPs) was elevated in metabolically active WAT depots, suggesting that HSPs were involved in lipolysis in WAT during cancer cachexia.

Nutritional support has been advocated as an adjunctive measure for several underlying treatments, including surgery and medical oncotherapy [10]. Omega-3 polyunsaturated fatty acids (PUFAs), especially docosahexaenoic acid and eicosapentaenoic acid, which are abundant in fish oil, can be beneficial to cancer treatment through maintaining a desired nutritional status [11] and decreasing the level of inflammatory factors [12], thereby increasing an overall survival rate [13]. PUFAs may be in a position to reduce several cancer-associated complications by regulating inflammatory pathways, serving as agonists for G protein-coupled receptors and/or altering the structure of cell membranes [14].

Recently, a randomized, triple-blind, placebo-controlled clinical trial showed that providing fish oil supplements to patients undergoing chemotherapy for gastrointestinal cancers may produce a favorable condition with no apparent treatment-related toxicity [15].

Selenium is an essential micronutrient implicated in many biological processes. Selenium displays anticancer properties by regulating expression of redox-active proteins as well as modulating a favorable intracellular redox status [16]. Selenium can be divided into inorganic and organic selenium. Inorganic selenium mainly comprises selenate and selenite that feature somewhat low bioavailability and toxicity [17], whereas organic selenium (such as selenium in yeast) provides antioxidant benefits as an antioxidant or a source of selenium for the synthesis of selenium-dependent antioxidants or proteins [18]. An animal study was recently performed, in which pregnant sows were assigned to three different groups and given various forms of selenium supplements over a 90-day gestation period. The experimental result revealed that only organic selenium was successfully reducing the expression of IL-1β, IL-6, and NF-κB in the ileum of newborn piglets. Moreover, the group receiving organic selenium showed a reduced level of ileal p-NF-κB and Beclin-1 proteins when compared to both the control group and the group given inorganic selenium. Clearly, the organic selenium was shown to be more effective in controlling both chronic and acute inflammation [19].

NuF is a total nutrition formula rich with energy, proteins, fish oil, selenium yeast, vitamins, and micronutrients. We have previously demonstrated that NuF helps maintain stable body weight and the level of serum albumin and prealbumin in patients undergoing radiotherapy [20]. However, the impacts of NuF, Tarceva, or in combination on anticancer and/or metabolic dysfunction were not addressed before. Lewis lung carcinoma (LLC) cells, which naturally overexpress endogenous EGFR, form tumors at a fast clip and result in cachexia in syngeneic C57BL/6 mice. In this study, we employed a murine lung-cancer-cachexia model to illustrate how NuF ameliorates tumor-induced AT wasting and improves cancer cachexia by preserving adipose mass.

## 2. Materials and Methods

### 2.1. Cell Line and Animal

LLC cells (CRL-1642) were obtained from the American Type Culture Collection (Manassas, VA, USA). Cells were cultured in Dulbecco’s modified Eagle medium (Sigma-Aldrich, St. Louis, MO, USA) supplemented with 10% fetal bovine serum (Sigma-Aldrich, USA), 2 mmol/L L-glutamine, 100 U/mL penicillin, and 100 mg/mL streptomycin. Cultures were maintained at 37 °C in a humidified atmosphere of 5% CO_2_ and passaged every 2 days.

Male C57BL/6 mice (4 weeks old) were obtained from the National Laboratory Animal Center. Mice were individually housed in a climate-controlled room (12:12 h dark–light cycle with a constant room temperature of 21 °C ± 1 °C). Mice were given at least 7 days to adjust to their new environment and diet before treatment. Mice were given free access to water and food (laboratory rodent diet, LabDiet 5001, Hubbard, OR, USA). After acclimatization, mice were divided into weight-matched groups.

### 2.2. Cell Viability Assay

For the cell viability study, 1 × 10^4^ cells resuspended in 100 µL medium were added per well to a 96-well plate for 24 h. LLC cells were then treated with 0, 0.625, 1.25, 2.5, 5, and 10 µM Tarceva for 24 h. Cell proliferation was monitored by using the Cell Titer 96 Aqueous One Solution Assay (Promega Corporation, Madison, WI, USA). Briefly, after the treatment, 20 mL of the combined 3-(4,5-dimethylthiazol-2-yl)-5-(3-carboxymethoxyphenyl)-2-(4-sulfophenyl)-2H-tetrazolium inner salt (MTS; Abcam, Cambridge, UK) and phenazine methosulfate solution was added into wells containing 100 mL of culture medium, incubated for 40 min at 37 °C in a humidified, 5% CO_2_ atmosphere. The absorbance at 490 nm was then measured using an ELISA reader.

### 2.3. Experimental Diets

NuF is a complete nutrition supplement (3.32 kcal/mL, 294 kcal/75 g; proteins, 22% of calories; lipids, 26% of calories; and carbohydrates, 50% of calories) that is rich in fish oil (20 mg/g NuF), selenium yeast (1.47 µg/g NuF), and CoQ10 (0.4 mg/g NuF). NuF was obtained from New Health Enterprise Inc. (Tustin, CA, USA).

### 2.4. Experimental Design

Mice were inoculated subcutaneously with a homogenate of tumor cells (3 × 10^5^) on day 0 (six mice per group, Figure 1A). The control group (NT group) was injected with 0.1 mL of sterile saline solution. After 7 days, tumor-bearing mice were distributed into four groups as follows: (1) no treatment group (T group), (2) receiving Tarceva 2 mg/kg/day (TT group), (3) receiving NuF 30 g/kg/day (TN group), and (4) receiving both Tarceva and NuF (TTN group). Tarceva and NuF were provided via oral gavage feeding six times each week for 3 weeks. Following the inoculation of tumor cells or PBS, body mass, food intake and tumor size were measured five times a week. Tumor growth was assessed by caliper measurement of two bisecting diameters in each tumor, and the tumor volume was calculated using the following equation: tumor volume (mm^3^) = width × length^2^/2. Animals were euthanized by CO_2_ inhalation on day 28, and the organs were removed, weighed, and then stored at −20 °C. Experimental procedures were approved by the Animal Ethical Committee and followed the principles of good laboratory animal care. Animal experiments complied with the guidelines for the maintenance and handling of experimental animals established by the Institutional Animal Care and Use Committee (IACUC) of the Hung-Kuang University of College of Medical and Health Care (HK105-101). 

### 2.5. Tissue Collection and the Levels of Serum Albumin

Skeletal muscles (gastrocnemius muscle), WAT, BAT [2], tumors, and lungs were dissected and weighed. Carcass weight was calculated by subtracting tumor weight from the body weight. Final body weight gain was calculated as the difference between the carcass and initial weight during the experimental periods. The level of serum albumin was measured for experimental mice using the SPOTCHEM EZ SP-4430 dry chemical system (Arkray, Kyoto, Japan).

### 2.6. Histopathological Analysis

Excised WAT was fixed in 10% formalin for 48 h at room temperature. Fixed WAT was trimmed into an appropriate size and shape and placed in embedding cassettes. Samples were dehydrated in a series of ethanol dilutions, passed through xylene and xylene/paraffin, and finally embedded in paraffin. Paraffin sections were prepared as 5 μm slices and placed on glass slides coated with Vectabound reagent (Vector Laboratories, Burlingame, CA, USA). For staining, slides were dipped in xylene to remove paraffin twice for 10 min, and then xylene was removed with a graded alcohol series (100%, 95%, and 70%). Slides were then washed with deionized water for 5 min. Sections were stained with Giemsa stain diluted 1:20 with deionized water for 1.5 h. Sections were washed in 0.5% acetic acid solution twice and then dehydrated in isopropyl alcohol and xylene. Microscopic observations were performed at 10× and 40× magnifications.

### 2.7. RNA Extraction and RT-qPCR

Total RNA was extracted from tumor tissues with a commercial RNA mini kit (Qiagen, Hidlen, Germany). The extracted RNA was then converted into corresponding cDNA using the M-MLV reverse transcriptase and an oligo-dT15 primer (Promega, Madison, WI, USA). The reverse transcription was performed at 37 °C for 60 min, and the reaction enzyme was then inactivated by heating at 70 °C for 5 min. Primers for real-time qPCR were designed with the Primer3 software (version 4.1.0), and DNA products were confirmed via electrophoresis. Reactions were conducted using the Bio-Rad iCycler iQ system and Sybr-Green PCR mix (iCycler iQ Real-Time PCR Detection System, Bio-Rad, Hercules, CA, USA). The PCR protocol was set as follows: initial denaturation at 95 °C for 3 min, 40 cycles of denaturation at 95 °C for 5 s, annealing at 60 °C for 1 min, and extension at 72 °C for 30 s. Relative gene expression was calculated using the comparative threshold cycle (Ct) method [21]. To quantify gene expression, the expression levels of the target genes were normalized to the endogenous reference gene, Gapdh. The relative quantity of each target gene, compared to a calibrator (normal pooled expression), was determined using the formula 2^−ΔΔCT^. Here, ΔCT represents the difference between the CT value of the target gene and the CT value of GAPDH, while ΔΔCT is the difference between the ΔCT of any given sample and the ΔCT of the calibrator [22]. RT-qPCR primer sequences were *Gapdh* forward, 5′-GCGACTTCAACAGCAACTC; *Gapdh* reverse, 5′-GGTCCAGGGTTTCTTACTCC; *Il6* forward, 5′-CCTCTGGTCTTCTGGAGTACC; *Il6* reverse, ACTCCTTCTGTGACTCCAGC; *PTHrP* forward, 5′-CAGCCGAAATCAGAGCTACC; *PTHrP* reverse, 5′-CTCCTGTTCTCTGCGTTTCC; *Ucp1* forward, 5′-CGACTCAGTCCAAGAGTACTTCTCTTC; *Ucp1* reverse, 5′-GCCGGCTGAGATCTTGTTTC; *ATGL* forward, 5′-TGTGGCCTCATTCCTCCTAC; *ATGL* reverse, 5′-TCGTGGATGTTGGTGGAGCT; *HSL* forward, 5′-GCTGGGCTGTCAAGCACTGT; *HSL* reverse, 5′-GTAACTGGGTAGGCTGCCAT.

### 2.8. Protein Extraction and Western Blotting

In brief, the tumor samples were sonicated at 4 °C in 1000 μL of a buffer solution containing 20 mM Tris-HCl (pH 7.5), 2 mM ATP, 5 mM MgCl_2_, 1 mM dithiothreitol (DTT), and 5 μL of a protease inhibitor cocktail (Sigma). The mixture was centrifuged at 10,000× *g* for 30 min, after which the supernatants were collected. Protein concentration was measured using the BCA™ Protein Assay Kit (Thermo Fisher Scientific, Waltham, MA, USA). Proteins (either 20 or 50 μg) were denatured by heating in a sample loading buffer [50 mM Tris-HCl (pH 6.8), 100 mM DTT, 2% SDS, 0.1% bromphenol blue, and 10% glycerol], separated by SDS-PAGE on a 12.5% polyacrylamide gel with 0.1% SDS, and then transferred onto immobilon polyvinylidene difluoride (PVDF) membranes (Millipore, Burlington, MA, USA). The membranes were blocked using 5% skim milk with 0.1% Tween-20 and subsequently incubated with polyclonal antibodies. Proteins were detected using antibodies against mouse EGFR (Santa Cruz Biotechnology, Dallas, TX, USA), pEGFR (Santa Cruz Biotechnology, USA), IL-6 (Abcam, UK), PTHrP (Santa Cruz Biotechnology, USA, HSP72 (Thermo Fisher Scientific, USA), total HSP25 (Thermo Fisher Scientific, USA), pHSL (Cell Signaling Technology, Danvers, MA, USA), HSL (Cell Signaling Technology, USA), ATGL (Abcam, UK), and UCP-1 (Santa Cruz Biotechnology, USA). Antibodies were then stripped off the membrane and reprobed with a specific antibody against β-actin (Novus Biologicals, Centennial, CO, USA). The intensity was quantified using the Fotodyne Image Analysis System (Fotodyne, Hartland, WI, USA) and the TotalLab software version 1.0 (Nonlinear Dynamics, Durham, NC, USA).

### 2.9. Statistical Analysis

The data were presented as the mean ± standard deviation. Statistical significance was assessed using the one-way ANOVA followed by the Bonferroni’s multiple comparison test (Prism Graph Pad 9). A *p*-value of less than 0.05 was considered statistically significant.

## 3. Results

### 3.1. A Special Nutritional Formula (NuF) Enhanced the Antitumor Effects of Tarceva

This study sought to investigate whether the total nutrition supplement enriched with fish oil and selenium yeast (NuF) could assist in the efficacy of the TKI drug, Tarceva, and mitigate cancer cachexia. Tarceva is known to be effective when EGFR is overexpressed or amplified. Previous studies have shown that LLC/LL2 lung cancer cells exhibit a high expression level of EGFR in several cancer cell lines [23]. We first confirmed that Tarceva significantly inhibited LLC cell growth after treatment with 1.25 µM for 24 h (Figure 1B) in a dose-dependent manner, indicating that LLC is a suitable model for investigating whether NuF enhances the effectiveness of Tarceva.

LLC cells were subcutaneously injected into C57BL/6 mice, followed by oral administration of Tarceva and/or NuF, either individually or in combination, six times per week until euthanasia on day 28. Tarceva and NuF treatments significantly reduced tumor weight. NuF exhibited greater variability than the one without NuF likely due to a tumor-suppressive effect in its own right. Tarceva demonstrated a typical inhibition activity (Figure 1C). In general, tumors were all >5 g in the Tarceva group; however, when Tarceva and NuF were administered together, the effect of the tumor growth inhibition was amplified to the greatest extent. Specifically, 78% of mice showed a reduced tumor weight to 3–5 g, and 22% of mice had <3 g tumors, indicating that NuF enhances the efficacy of Tarceva in tumor suppression (Figure 1D). It is clear that Tarceva effectively inhibited tumor metastasis to the lungs and reduced the expression of phosphorylated EGFR (pEGFR) in tumors. Interestingly, we found that NuF on its own also reduced tumor pEGFR expression and the number of lung tumor metastases (Figure 1E,F). When both agents were combined, NuF did not diminish the efficacy of Tarceva; the combination, in fact, elevated the inhibition of tumor growth and metastasis when compared with either agent alone.

### 3.2. Combination of Tarceva and NuF Prevented Cachexia Characteristics

Next, we investigated the effect of a given nutritional combination on the parameters of cachexia diagnostic items, typically including losses of weight, lean body and fat mass, and abnormalities in levels of albumin. Table 1 demonstrates that there are no significant differences in the initial weight of mice among all groups. However, except the mice treated with both Tarceva and NuF (TTN group), the carcass weight of the tumor-bearing mice in other groups was significantly lower than that of the normal mice (NT group) after the experiment. NuF was administered once a day via gavage at a dose of 30 g/kg, while the rest of the time, it was offered through ad libitum feeding. The food intake, total calorie intake, and food efficiency ratio (FER) of the mice were recorded daily. It is clear that Tarceva and NuF can mitigate tumor-induced body weight loss. When compared with the normal mice, the tumor-bearing mice showed a significant decrease in FER, indicating that tumors led to a decreased food efficiency. The FER of mice makes no significant difference among the TTN groups, while there is a trend toward minor increased FER (Table 1).

All tumor-bearing mice showed significantly lower levels of serum albumin than the NT group. In contrast, the albumin level of the mice treated with Tarceva and NuF was significantly higher than that of the NT group (Figure 2A). Next, the gastrocnemius muscle (Gastroc), which was selected as a representative of skeletal muscles, and the epididymal fat (WAT) and interscapular brown AT (BAT) which were selected as representatives of Ats, were examined (Figure 2B,C). All tumor-bearing mice experienced losses of Gastroc, WAT, and BAT masses, underlining tissue losses as a result of cancer cachexia. Neither Tarceva nor NuF could attenuate tumor-induced muscle atrophy. Tumor-bearing mice treated with NuF, however, had significantly higher WAT mass than Tarceva groups (T: 0.09 ± 0.05 vs. TT: 0.12 ± 0.04 vs. TN: 0.22 ± 0.04 vs. TTN: 0.24 ± 0.04). Additionally, AT was preserved to a significant extent in the group receiving the combined treatment with Tarceva and NuF. Pathological examination with hematoxylin and eosin (H&E) staining further revealed white adipose atrophy and depletion of fat depots in the T group (Figure 2D). However, NuF alone or in combination with Tarceva showed remarkable lipid accumulation, indicating improvement in the adipose atrophy.

### 3.3. Tarceva and NuF Acted Synergistically in Suppressing Adipocyte Dysfunction Factor in the Tumor Microenvironment

We next analyzed the protein and gene expression levels of IL-6 and PTHrP in tumors because low food efficiency and AT depletion were observed in tumor-bearing mice and as IL-6 and PTHrP, known as tumorkines, are mediators in fat depletion. As shown in Figure 3, the treatment with Tarceva or NuF could inhibit tumor IL-6 protein expression, whereas the NuF treatment could inhibit the PTHrP expression at both the protein and gene levels (Figure 3A,B).

Manisterski et al. reported that the activation of the hypoxia-inducible factor (HIF) promotes the secretion of PTHrP, which preconditions an appropriate bone marrow microenvironment for breast cancer colonization [24]. Therefore, we investigated the effect of an individual administration of Tarceva or NuF and their combination on the expression of HIF-1 subunits, HIF-1α, and HIF-1β in tumors. Tarceva alone could inhibit tumor HIF-1α protein expression but with no change in regard to the HIF-2α expression. NuF had no effect on tumor HIF-1α expression but displayed a somewhat cooperative inhibitory effect alongside Tarceva on the HIF-1α expression. Interestingly, the NuF treatment significantly reduced the tumor HIF-2α expression (Figure 3C,D). These results suggest that Tarceva and NuF exert different modes of action on the HIF-mediated biological functions. 

The expression of PTHrP made no significant difference between the TT and T groups at the gene and protein levels. The treatment with NuF, however, decreased the expression of PTHrP. The reduction in the PTHrP expression in tumors can be attributed to the NuF-mediated inhibition on HIF-2α as well as the Tarceva-mediated inhibition on HIF-1α as manifested from the TTN group.

### 3.4. NuF Prevented Tumor-Induced WAT Browning

Given that NuF effectively inhibits the expression of PTHrP and ameliorates fat wasting (Figure 2B) and that PTHrP is a mediator of AT browning and tumor-induced cachexia, the protein and mRNA levels of lipolytic enzymes and thermogenic effectors involved in the browning process needed to be known in the first place (Figure 4A,B). The expression of the *Il6* gene made no change amid the adipocytes isolated from these groups. However, the thermogenic effectors were indeed increased in all tumor-bearing mice at both the mRNA and protein levels, for example, thermogenic factor UCP-1 in the white fat depots. The LLC tumors also induced several lipolytic proteins, such as adipocyte triglyceride lipase (ATGL) and phosphorylated hormone-sensitive lipase (HSL), while there were no differences concerning both the total HSL protein and the gene expression level at the selected WAT depots between the sham- and tumor-treated animals. The treatment with NuF significantly inhibited the mRNA expression of *Ucp1* and *ATGL* as well as the HSL activity. When NuF was provided together with Tarceva, both thermogenic and lipolytic activities were decreased to a significant extent. Added together, these results suggest that NuF is in a position to block the PTHrP signaling as well as to inhibit adipose tissue browning and shrinking thus underscoring a synergistic suppression effect as a result of NuF in conjunction with Tarceva.

The expression level of the heat shock protein (HSP) is also related to the overall metabolic activity of AT. Rogers et al. reported that the expression of HSP72 and HSP25 was positively correlated to an elevated metabolic activity [25]. To better know the intrinsic connection between LLC tumors-induced AT wasting and HSP, the elevated expression levels of HSP72 and HSP25 in WAT (Figure 4C) of the tumor-bearing mice versus that of control mice support the assumption. That is, NuF significantly decreased the HSP72 expression and lowered the HSP25 level to a relatively mild extent in WAT when compared with those in T or TT mice. As a result, NuF is able to preserve AT by inhibiting the expression of HSP72 in WAT and the correlated lipolysis in tumor-bearing mice. 

## 4. Discussion

Cancer cachexia is an ensemble manifestation of a multifaceted metabolic disorder that often hinders the effectiveness of standard cancer treatments. This fact, however, underscores that adequate nutritional support is able to meet the unmet pathophysiological demands of cancer cachexia. The ESPEN guidelines have emphasized the need by recommending an integrated approach that includes nutrition, physical activity, and medication for managing cancer cachexia [26]. Providing high-quality nutritional support is essential for cancer patients, as preventing the onset of cancer cachexia is a key aspect of routine clinical care. Numerous studies have independently indicated that nearly half of cancer patients experience substantial weight loss and increased resting energy expenditure [27]. Recent research has further uncovered a notable shift in energy metabolism of cancer patients, highlighting the critical role of fat browning in this process [8]. Thus, identifying favorable nutritional entities that slow down fat browning and AT breakdown ought to be beneficial to cancer patients. Lung adenocarcinoma, the primary cause of cancer-related deaths globally, accounts for nearly 80% of NSCLC cases [28]. Treatment options for advanced NSCLCs include chemotherapy, radiotherapy, and EGFR-TKI therapy. Activation of the EGFR-tyrosine-kinase downstream signaling fosters a malignant phenotype leading to the stimulation of crucial oncogenic pathways in reference to cell survival, proliferation, invasion, and metastasis [29]. The EGFR overexpression is especially prevalent in squamous tumors and correlates well with a more aggressive phenotype and poorer prognosis [30]. It has been known that LLC cells overexpress EGFR and that Tarceva prohibits LLC growth as well as reduces EGFR activation in tumors, thereby preventing its metastasis to lungs (Figure 1). Our previous reports have shown that fish oil and selenium yeast in NuF can impede tumor growth and subsequent metastasis by the inhibition of angiogenesis and induction of apoptosis [31]. In this study, NuF is further revealed in a position to inhibit EGFR phosphorylation in tumors (Figure 1).

Hypoxia is another crucial factor in solid tumors, which promotes malignant progression and drug resistance. HIF, a heterodimeric complex composed of a regulated subunit (e.g., HIF-1α and HIF-2α) and a constitutive subunit (HIF-1β), is a major regulatory protein facilitating cancer cell colonization [32]. EGFR has been implicated as a hypoxia-independent driver for HIF expression. EGFR activation is a critical switch to regulate HIF transcription and translation in cells [33]. HIF-1α and HIF-2α have distinct expression patterns and functions [34], which play important roles in cancer development [34]. Pore et al. reported that the reduction in the VEGF expression in head and neck cancer after the Tarceva treatment can be attributed to the inhibition of HIF-1α [33]. By the same token, our results showed that Tarceva reduces the expression of HIF-1α in tumors rather than that of HIF-2α (Figure 3). In contrast, NuF inhibits HIF-2α expression. The combination of Tarceva and NuF unexpectedly maximized the anticancer activity through inhibiting both HIF-1α and HIF-2α in a synergic manner, thereby outperforming other treatments with the best effectiveness. 

Despite new advancements in diagnosis and therapy, the prognosis and survival rates for patients with lung cancer remain unsatisfactory [35]. Cancer cachexia, a common concern in advanced lung cancer patients, is the key factor that leads to poor treatment tolerance, prognosis, and survival outcomes [36]. Approximately 60% of patients with lung cancer experience intrinsic weight loss, and >10% succumb to cancer cachexia [36]. Complications of cachexia and sarcopenia that are the major determinant with respect to the morbidity and mortality of NSCLC [37] indeed impair treatment responses. An effective therapy for cancer cachexia remains limited, while the effectiveness of NuF on cachexia has shed new light on this matter. 

A study with a cohort of 311 cancer patients revealed that the primary cause of weight loss is fat loss, in which the trunk is affected first, followed by legs and arms [38]. In cachexia, the loss of fat is more severe than the loss of muscle, suggesting an unmet need to mitigate fat loss [38]. Nevertheless, the mitigation of body weight loss in tumor-bearing mice in the present study is coincident with a reduction in fat loss alongside muscle wasting (Figure 1) with the administration of NuF, which keeps the weight of AT (especially WAT) and lipid droplets within a normal scope (Figure 1D). 

A recent study demonstrated that WAT “browning” confers some thermogenic properties to the adipocytes, leading to wasteful energy expenditure [39]. During cancer cachexia, catabolism is driven by some cytokines, such as IL-6, and tumorigenic factors, such as PTHrP. Since NuF can stabilize the weight of BAT and WAT, we were keen to know whether NuF could also prevent AT from browning. Our data suggest that NuF is indeed able to inhibit PTHrP and gene expression, in addition to minor suppression of IL-6 expression. Since hypoxia can induce *PTHrP* gene transcription via HIF-2α [24], we hypothesized that NuF follows suit in reducing tumor PTHrP expression by downregulating HIF-2α (Figure 3). 

Next, we extracted epididymal fat (WAT) from mice to count the levels of UCP1 (a marker for brown fat activation) and ATGL and HSL (two fat-degrading enzymes). As a result, NuF is in a position to downregulate the gene expression level of *Ucp1* and *ATGL* in WAT as opposed to that of *HSL* that was not much altered despite a downward trend observed (Figure 4). Given that HSPs take part in AT degradation [25], NuF was further shown to regulate expression of HSP72, thus suggesting that it may also inhibit WAT lipolysis through suppressing the expression of HSP72 (Figure 4C). 

Though the current conclusion is limited to rodents, its extension to human can be expected in spite of the need for clinical validation in man. At the current stage, the combination protocol potentiates its translation to act as an adjuvant for patients receiving gastrostomy procedures and/or chemotherapy of typical oncologic treatment [40].

## 5. Conclusions

NuF is confirmed to be able to inhibit tumor EGFR activation and the expression of HIF-2α and PTHrP. NuF is also able to suppress the expression of UCP-1 and lipolytic enzymes, preventing WAT from browning. Moreover, NuF can inhibit the function of HSP72, thereby maintaining AT integrity. Added together, NuF supplements and/or complements Tarceva by enhancing drug efficacy as well as modulating fat-loss associated cancer cachexia (Figure 4D). Therefore, the combination of NuF with Tarceva exerts an extraordinary synergistic effect to slow down tumor progression and prevent the loss of adipose tissue.

## Figures and Tables

**Figure 1 cells-13-01485-f001:**
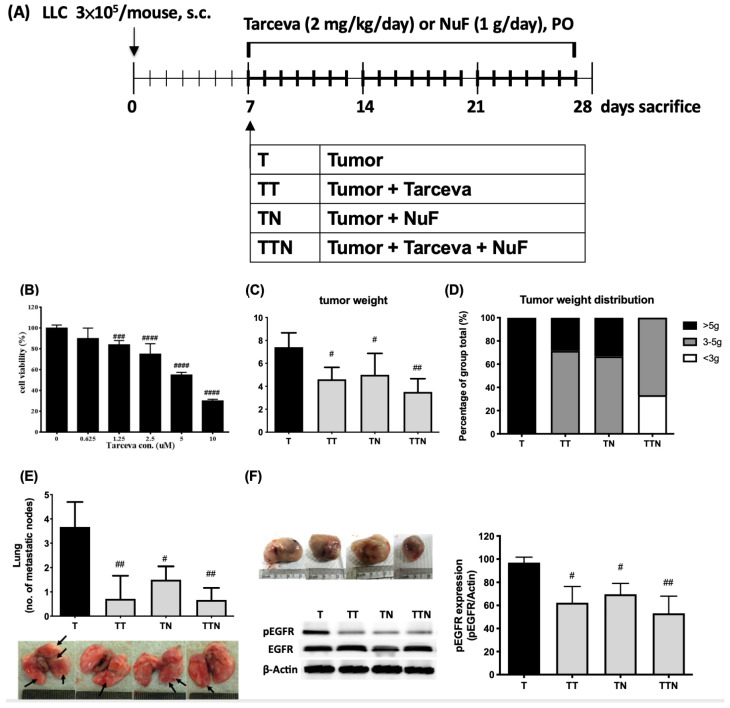
Anticancer and antimetastatic effect of the combination of Tarveva and NuF in tumor-bearing mice. (**A**). The study involved a treatment regimen combining Tarceva with NuF in mice with tumors. On day 7, Lewis lung carcinoma (LLC) cells (3 × 10^5^) were injected subcutaneously into the right dorsal side of C57BL/6 mice. Tumor volume was measured using the following formula: 1/2 (x^2y), where x represents tumor width and y represents tumor length. Tumor-bearing mice were randomly assigned to four groups as follows: the control group with no treatment (T), the TT group (Tarceva at 2 mg/kg/day), the TN group (NuF at 1 g/mouse/day), and the TTN group (Tarceva at 2 mg/kg/day combined with NuF at 1 g/mouse/day). After 28 days, mice were sacrificed, and tumors, gastrocnemius muscles, white adipose tissue, brown adipose tissue (BAT), and lungs were collected for further analysis. (**B**). MTS assay for determining the inhibition of LLC cell growth by Tarveva. (**C**). Tumor weight. Results are based on three independent replicates. (**D**). Tumor weight distribution. (**E**). The average number of lung metastatic nodules. Representative photos of the lungs; arrows point to the metastatic nodules. (**F**). The image on the left displays the expression level of EGFR and its phosphorylated form in tumors from each group. Meanwhile, the image on the right illustrates the quantified ratio of phosphorylated EGFR to total EGFR (pEGFR/total EGFR) following treatment with Tarceva and NuF. ^#^ *p* < 0.05, ^##^ *p* < 0.01, ^###^ *p* < 0.001, and ^####^ *p* < 0.0001 compared to the T group. Data are expressed as means ± SD. N = 5–6 samples per group. Each group consisted of 5 to 6 mice.

**Figure 2 cells-13-01485-f002:**
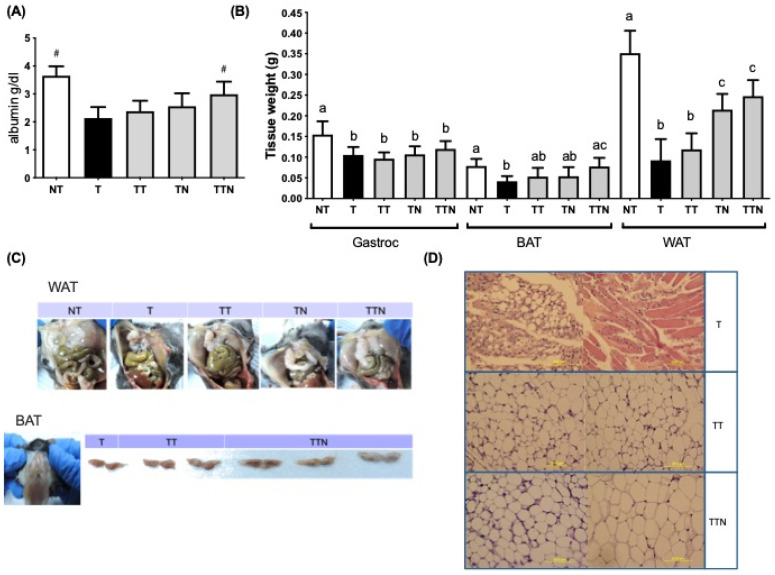
Anticachexic effect of the combination of Tarveva and NuF in tumor-bearing mice. (**A**) Albumin level. (**B**) Gastrocnemius muscle (Gastroc), epididymal fat (WAT) and interscapular brown adipose tissue (BAT) weight. (**C**) Image of BAT. (**D**) H&E staining image of WAT. Scale bar = 200 µm. ^#^ *p* < 0.05 as compared to the T group. Different letters in the groups represent significant differences. Data are expressed as means ± SD. N = 5–6 samples per group.

**Figure 3 cells-13-01485-f003:**
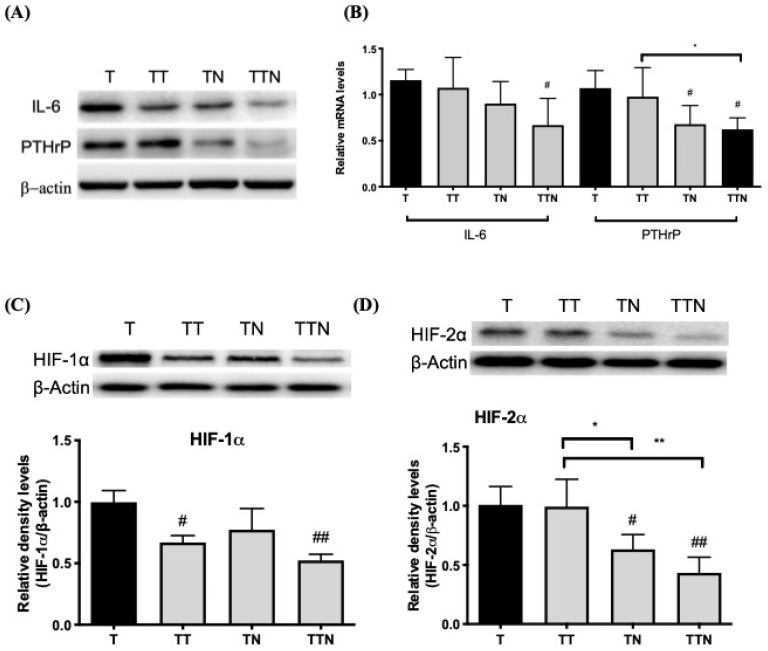
Co-administration of Tarveva and NuF inhibited adipocyte dysfunction factor in tumors. (**A**) Representative Western blots of IL-6, PTHrP, and β-actin in tumors from LLC tumor-bearing mice. (**B**) Relative mRNA expression levels of *Il6* and *PTHrP* were measured via RT-qPCR. Values are means of fluorescence signals expressed as a percentage of no-treatment tumor mice (T), and normalization to the *Gapdh* mRNA. (**C**) Western blot analysis for the expression of HIF-1α and β-actin in tumors. The graph represents the relative densitometric intensity of each band normalized to β-actin. (**D**) Western blot analysis for the expression of HIF-2α and β-actin in tumors. ^#^ *p* < 0.05 and ^##^ *p* < 0.01 as compared to the T group. * *p* < 0.05 and ** *p* < 0.01 as compared between two groups. Data are expressed as means ± SD. N = 5–6 samples per group.

**Figure 4 cells-13-01485-f004:**
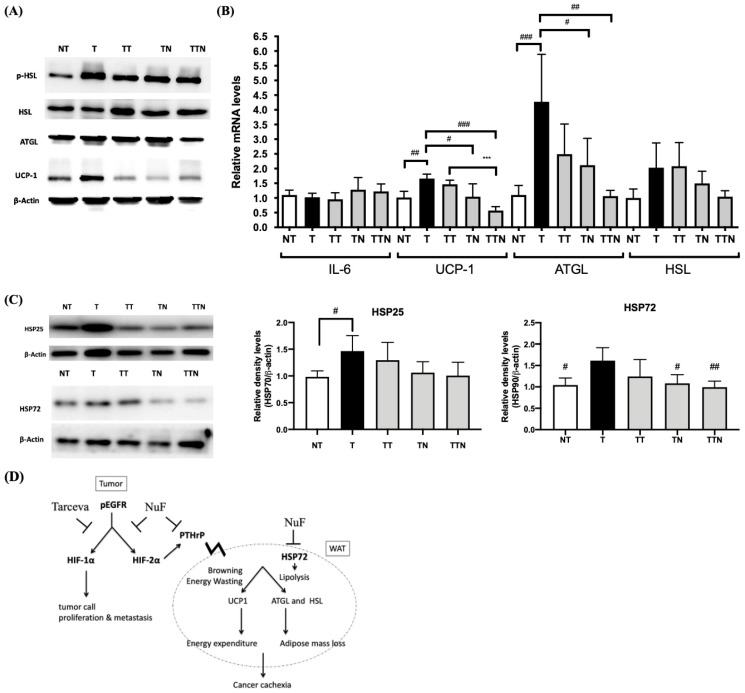
NuF suppresses the expression of thermogenic and lipolytic factors in white adipose tissue (WAT). (**A**). Representative Western blots of pHSL, total HSL, UCP-1, and β-actin in WAT from LLC tumor-bearing mice. (**B**). Relative mRNA expression levels of *Il6*, *Ucp1*, *Argl*, and *HSL* were measured via RT-qPCR. Values are means of fluorescence signals expressed as a percentage of health control mice (NT group), and normalization to the *Gapdh* mRNA. (**C**). A representative Western blot of HSP25, HSP72, and β-actin expression in WAT. The graph represents the relative densitometric analysis of each band normalized to β-actin. (**D**). Diagram showing the hypothesized underlying mechanism for NuF inhibition of tumor progression and adipose tissue atrophy. UCP1, uncoupling protein 1. ATGL, adipocyte triglyceride lipase. HSL, hormone-sensitive lipase. HSP, Heat shock protein. ^#^ *p* < 0.05, ^##^ *p* < 0.01, and ^###^ *p* < 0.001 as compared to the T group. *** *p* < 0.001 as compared between the two groups. Data are expressed as means ± SD. N = 5–6 samples per group.

**Table 1 cells-13-01485-t001:** Effect of Tarceva plus NuF on body weight and food intake.

(A) Body weight, weight gain, diet intake and food efficiency ratio (FER)							
	Initial weight (g)	Carcass weight (g)		Weight gain (g/4 weeks)	Diet intake (g/4 weeks)		FER
NT	20.84 ± 0.75	23.77 ± 1.23 ^a^		2.93 ± 1.29 ^a^	88.2 ± 9.69		3.33 ± 1.47 ^a^
T	22.11 ± 1.44	20.31 ± 1.81 ^b^		−1.80 ± 1.99 ^b^	90.1 ± 8.10		−2.00 ± 2.21 ^b^
TT	22.01 ± 1.55	20.79 ± 1.48 ^b^		−1.22 ± 1.05 ^b^	87.7 ± 8.10		−1.40 ± 1.19 ^b^
TN	21.97 ± 0.94	19.78 ± 1.95 ^b^		−0.99 ± 0.75 ^b^	90.4 ± 9.50		−1.09 ± 0.83 ^b^
TTN	21.92 ± 1.54	21.04 ± 1.38 ^a,b^		−0.88 ± 1.42 ^b^	93.1 ± 7.60		−0.94 ± 1.53 ^b^
(B) Food intakes (kcal per day)							
	Day10	Day13	Day15	Day17	Day21	Day23	Day26
NT	14.61	12.52	13.03	14.23	13.59	11.36	10.83
T	14.43	13.96	12.48	13.66	14.22	11.60	11.79
TT	15.13	13.04	12.92	12.79	12.47	11.95	11.36
TN	14.66	14.63	13.05	13.20	13.38	10.56	12.62
TTN	14.80	14.15	12.88	14.35	14.35	12.10	12.27

NT = non-tumor mice, T = tumor-bearing mice, TT = tumor-bearing mice receiving Tarceva, TN = tumor-bearing mice receiving NuF, TTN = tumor-bearing mice receiving Tarceva and NuF combination. FER = weight gain/food intake for experiment. Data represent mean ± SD. Different letters represent significant differences at *p* < 0.05 probability level. Each group consisted of 5 to 6 mice.

## Data Availability

The original contributions presented in the study are included in the article, further inquiries can be directed to the corresponding authors.

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
