# Peer review of "Supplementation with Fish Oil and Selenium Protects Lipolytic and Thermogenic Depletion of Adipose in Cachectic Mice Treated with an EGFR Inhibitor"

_cells, 2024, doi:10.3390/cells13171485_

Round 1

Reviewer 1 Report

Comments and Suggestions for Authors

Line 44-45 what is the role of Tarceva on epidermal growth factor receptor (EGFR)? agonist or antagonist.

Line 77 could you add information about de possibles mechanisms of cell signaling of the Omega-3 PUFAs for cancer treatment.

Line 92-93 What kind of inflammation is improved with organic selenium supplementation, low grade inflammation or which? could you add more information about this question.

Line 102-105 You said that "we aimed to determine how NuF and Tarceva influence tumour progression and the ability to preserve adipose mass in the LLC model" is important add information in your conclusions that support the objective of this study.

Line 134 (2.4) you wrote “Tumor implantation and treatment” could be change this information by "experimental design".

Line 171-176 can you add major description about the three steps in PCR; RNA extraction, cDNA synthesis (add synthesis conditions) and qPCR conditions?

Line 179-180 could you cite the double delta Ct method (2△△CT).

Line 190-201 It requires a detailed description of protein extraction, as well as polyacrylamide gel electrophoresis, incubation with primary and secondary antibody, and membrane development.

203-205 In the statistical analysis of this study, did you use standard deviation or standard error? Which is recommended for this type of study according to the sample size?

Line 218-232 The protein expression of pEGFR is not described in the caption of Figure 1 and the figure has no letter (F), you need add more information about the protein expression because is loss.

Line 256-257 you said "additional 1 g per day" you need add per kg (g/kg) of weight mice.

Line 313 could you add information about the possible action mechanism of Tarceva and NuF?

Line 379-380 you said, “can reduce HIF protein translation in some cells” you can add “transcription and translation”.

Line 408-416 what is the role of IL-6 in this study, proinflammatory or anti-inflammatory?

It is important to add more information to support the results obtained in your study when compared with other studies, in the discussion section.

Line 432-433 “Hence, NuF is considered a potential therapeutic target for ameliorating cancer cachexia (Figure 4D)” this paragraph you need to change in discussion section.

Comments on the Quality of English Language

Moderate editing of English language required.

Author Response

Comment 1:

Line 44-45 what is the role of Tarceva on epidermal growth factor receptor (EGFR)? agonist or antagonist.

Response 1:

We thank the reviewer’s question and suggestion. I have incorporated the description into the article at Line 42.

『As an EGFR tyrosine kinase inhibitor, erlotinib (Tarceva) has demonstrated clinical efficacy by antagonizing the EGFR and is now approved for clinical use in various diseases.』

Comment 2:

Line 77 could you add information about de possibles mechanisms of cell signaling of the Omega-3 PUFAs for cancer treatment.

Response 2:

We thank the reviewer’s suggestion. I have incorporated the description into the article at Line 73.

Comment 3:

Line 92-93 What kind of inflammation is improved with organic selenium supplementation, low grade inflammation or which? could you add more information about this question.

Response 3:

We thank the reviewer’s question and suggestion. I have incorporated the description into the article at Line 87.

Comment 4:

Line 102-105 You said that "we aimed to determine how NuF and Tarceva influence tumour progression and the ability to preserve adipose mass in the LLC model" is important add information in your conclusions that support the objective of this study.

Response 4:

We thank the reviewer’s suggestion. I have incorporated the description into the article at Conclusions.

『Therefore, the combination of NuF with Tarceva can slow down tumor progression and prevent the loss of adipose tissue.』

Comment 5:

Line 134 (2.4) you wrote “Tumor implantation and treatment” could be change this information by "experimental design".

Response 5:

We thank the reviewer’s suggestion. The sentence has been rewritten.

Comment 6:

Line 171-176 can you add major description about the three steps in PCR; RNA extraction, cDNA synthesis (add synthesis conditions) and qPCR conditions?

Response 6:

We thank the reviewer’s suggestion. I have incorporated the description into the article.

Comment 7:

Line 179-180 could you cite the double delta Ct method (2△△CT).

Response 7:

We thank the reviewer’s suggestion. I have cite the double delta Ct method (2△△CT) into the article.

Comment 8:

Line 190-201 It requires a detailed description of protein extraction, as well as polyacrylamide gel electrophoresis, incubation with primary and secondary antibody, and membrane development.

Response 8:

We thank the reviewer’s suggestion. I have incorporated the description into the article.

Comment 9:

203-205 In the statistical analysis of this study, did you use standard deviation or standard error? Which is recommended for this type of study according to the sample size?

Response 9:

We thank the reviewer’s question. Standard deviation is utilized in our statistical presentation due to its capacity to directly represent the variability within the dataset.

Comment 10:

Line 218-232 The protein expression of pEGFR is not described in the caption of Figure 1 and the figure has no letter (F), you need add more information about the protein expression because is loss.

Response 10:

We thank the reviewer’s suggestion. I have incorporated the description into the article.

Comment 11:

Line 256-257 you said "additional 1 g per day" you need add per kg (g/kg) of weight mice.

Response 11:

We thank the reviewer’s suggestion. We have modified the units of measurement according to the reviewer's suggestion.

Comment 12:

Line 313 could you add information about the possible action mechanism of Tarceva and NuF?

Response 12:

We thank the reviewer’s suggestion. In Line 314, we describe the observed phenomenon. In the discussion section, we mentioned that previous studies have found that Tarceva reduces VEGF expression in head and neck cancer cells by inhibiting HIF-1α. Based on this, we also analyzed HIF-1α and confirmed that Tarceva indeed inhibits HIF-1α, consistent with other findings, whereas NuF showed no significant difference. Consequently, we also analyzed HIF-2α and unexpectedly discovered that NuF can inhibit HIF-2α expression.

Comment 13:

Line 379-380 you said, “can reduce HIF protein translation in some cells” you can add “transcription and translation”.

Response 13:

We thank the reviewer’s suggestion. The sentence has been rewritten.

Comment 14:

Line 408-416 what is the role of IL-6 in this study, proinflammatory or anti-inflammatory?

Response 14:

We thank the reviewer’s question. Given the focus of our study on tumors, IL-6 is considered a proinflammatory cytokine in this context.

Comment 15:

It is important to add more information to support the results obtained in your study when compared with other studies, in the discussion section.

Response 15:

We have expanded the discussion section in accordance with the reviewer's suggestions.

Comment 16:

Line 432-433 “Hence, NuF is considered a potential therapeutic target for ameliorating cancer cachexia (Figure 4D)” this paragraph you need to change in discussion section.

Response 16:

We thank the reviewer’s suggestion. The sentence has been rewritten.

Reviewer 2 Report

Comments and Suggestions for Authors

Wang et al studied the nutritional effects of fish oil and selenium in a mouse model of lung cancer. This is an establish Taiwanese group of investigators focusing on the nutritional/metabolic benefits of micronutrients. This present study investigates the mechanisms underscoring the adipose tissue wasting in the mouse model of lung cancer. Their results largely support their notion, i.e., their proposed nutrient capsule exerts beneficial effects on cancer cachexia-associated adipose tissue wastin, likely via the PTHrP/IL-6/HIF pathway.

The following aspects should improve the quality of this manuscript. 

Title – should be amended, it does not capture the essence of the study (general readers do not understand Tarceva)

Abstract – amendment is required, again, general readers do not know about Tarceva.

Introduction – sufficiently enlists the relevance of this current study, rationale, and literature review.

Methods – normally mean ± SD for clinical study and mean ± SEM is used for basic research.

Results 

Data presentation – whenever it is possible, please plot the raw data points instead of just using the bar throughout the entire manuscript (e.g., fig 1 B, C etc). What is the sample size for each group of mice and cell culture? By using the real raw data point will enable the readers to comprehensively see the experimental setting.

Font size is too small for certain data set (e.g., Fig 1).

Table 1 – sample size for each group is lacking.

For Table 1B, I presume all mice are fed ad libitum. Cancer mice are cachectic. But how come, the daily food intake of cancer mice, with or without treatment, are greater than normal mice through the study of 26 days? Reduced food intake is a hallmark for cachexia.

Fig 2D – detailed description of adipose tissue atrophy is lacking, can you co-stain UCP-1 for those white adipose tissue? What is the size of white adipocyte? And for comparison, the results from normal control mice should be included. 

Detailed molecular mechanism is lacking – authors should consider employing RNAseq to explore the adipose tissue metabolism. By doing some, they may find out some novel pathways underlying adipose tissue wasting.

Discussion

Quite a large amount of redundant information (as already listed in the section of discussion). The first paragraph should focus on the novelty and the important findings instead of literature review. 

Overall, the content of the discussion is very thin, authors need to incorporate relevant literature (on the mechanism of those important molecular implicated with cachexia/adipose tissue wasting/enhanced lipolysis). Enhanced lipolysis has been also associated with many other models of cachexia.

Comments on the Quality of English Language

Language editing is needed.

Author Response

Comment 1:

Title – should be amended, it does not capture the essence of the study (general readers do not understand Tarceva)

Response 1:

We thank the reviewer’s suggestion. We have made some slight revisions to the title.

Comment 2:

Abstract – amendment is required, again, general readers do not know about Tarceva.

Response 2:

We thank the reviewer’s suggestion. We have made some slight revisions to the Abstract.

Comment 3:

Methods – normally mean ± SD for clinical study and mean ± SEM is used for basic research. Response 3:

We thank the reviewer’s question. We use standard deviation in our statistical presentation primarily because it directly represents the variability within the dataset.

Comment 4:

Results

Data presentation – whenever it is possible, please plot the raw data points instead of just using the bar throughout the entire manuscript (e.g., fig 1 B, C etc). What is the sample size for each group of mice and cell culture? By using the real raw data point will enable the readers to comprehensively see the experimental setting.

Response 4:

We thank the reviewer’s suggestion. We have added the sample sizes for the mice and cell cultures to the manuscript.

Comment 5:

Font size is too small for certain data set (e.g., Fig 1).

Response 5:

We thank the reviewer’s suggestion. We have increased the resolution of Figure 1.

Comment 6:

Table 1 – sample size for each group is lacking.

Response 6:

We thank the reviewer’s suggestion. We have added the sample sizes for the mice to the manuscript.

Comment 7:

For Table 1B, I presume all mice are fed ad libitum. Cancer mice are cachectic. But how come, the daily food intake of cancer mice, with or without treatment, are greater than normal mice through the study of 26 days? Reduced food intake is a hallmark for cachexia.

Response 7:

We thank the reviewer’s question. Not all cancers result in anorexia. In fact, we did not observe changes in food intake in another lung cancer mouse model [1]. Additionally, since the treatment in this experiment is targeted therapy rather than chemotherapy, it is less likely to cause anorexia, which is more commonly associated with chemotherapeutic drugs. The absence of anorexia actually allows our experimental results to be free from the confounding effects of reduced food intake.

  1. Wang, H., et al., Reduction of splenic immunosuppressive cells and enhancement of anti-tumor immunity by synergy of fish oil and selenium yeast. PLoS One, 2013. 8(1): p. e52912.

Comment 8:

Fig 2D – detailed description of adipose tissue atrophy is lacking, can you co-stain UCP-1 for those white adipose tissue? What is the size of white adipocyte? And for comparison, the results from normal control mice should be included.

Response 8:

We thank the reviewer’s question. Regarding the description of Fig 2D, since all three groups involve adipose tissue from tumor-bearing mice, we compared the adipose tissue of mice treated with the Tarceva and NuF with tumor-bearing alon mice. Our intention was to provide a brief introduction to the severe fat loss observed in tumor-bearing mice. We apologize for not including UCP-1 co-staining. However, in Fig 4, we have analyzed the protein and gene expression of UCP-1 in adipose tissue, and the results are compared with those from normal control mice.

Comment 9:

Detailed molecular mechanism is lacking – authors should consider employing RNAseq to explore the adipose tissue metabolism. By doing some, they may find out some novel pathways underlying adipose tissue wasting.

Response 9:

We thank the reviewer’s suggestion. We are also very eager to further employ RNAseq to explore adipose tissue metabolism. We hope the reviewer will grant us the opportunity to publish, allowing us to continue investing more resources and effort into discovering novel pathways underlying adipose tissue wasting.

Comment 10:

Discussion

Quite a large amount of redundant information (as already listed in the section of discussion). The first paragraph should focus on the novelty and the important findings instead of literature review.

Overall, the content of the discussion is very thin, authors need to incorporate relevant literature (on the mechanism of those important molecular implicated with cachexia/adipose tissue wasting/enhanced lipolysis). Enhanced lipolysis has been also associated with many other models of cachexia.

Response 10:

We thank the reviewer’s suggestion. We have expanded the discussion section in accordance with the reviewer's suggestions.

Reviewer 3 Report

Comments and Suggestions for Authors

Dear Authors,

I read with great interest your manuscript about dietary supplements in mice models receiving Tarceva treatment.

However, there are some aspects that require your attention.

There are many abbreviations in the text, please provide a list of abbreviations at the end of the manuscript.

In Figure 2 you have microscopy images, you need to mention the magnification level and to insert reference bars. Please update the figure.

In the discussion section you need to expand on the future use of these findings. Such future use could be the translation of these findings in improving the adjuvant treatment of patients with gastrostomy procedures and chemotherapy during oncologic treatment. Reference this to the work by Anghel AG, Anghel I, Dumitru M, Cristian D, Burcos T. The use of gastrostomy procedures in HNC patients. Chirurgia (Bucur). 2013 May-Jun;108(3):341-5. PMID: 23790782.

Before the conclusions you need to insert a paragraph about the clear limitations of the present study.

In the reference list you have titles from 1974, please update the reference list as I am sure there are other newer articles on the subject.

Looking forward to receiving the improved version of your manuscript.

Author Response

Comment 1:

There are many abbreviations in the text, please provide a list of abbreviations at the end of the manuscript.

Response 1:

We thank the reviewer’s suggestion. We provide a list of abbreviations at the end of the manuscript.

Comment 2:

In Figure 2 you have microscopy images, you need to mention the magnification level and to insert reference bars. Please update the figure.

Response 2:

We thank the reviewer’s suggestion. We have added a scale bar in the lower-left corner of the microscopy images, indicating 200 µm.

Comment 3:

In the discussion section you need to expand on the future use of these findings. Such future use could be the translation of these findings in improving the adjuvant treatment of patients with gastrostomy procedures and chemotherapy during oncologic treatment. Reference this to the work by Anghel AG, Anghel I, Dumitru M, Cristian D, Burcos T. The use of gastrostomy procedures in HNC patients. Chirurgia (Bucur). 2013 May-Jun;108(3):341-5. PMID: 23790782. Response 3:

We thank the reviewer’s suggestion. We have followed the suggestion and cited this reference to expand on the potential future applications of our study's findings.

Comment 4:

Before the conclusions you need to insert a paragraph about the clear limitations of the present study.

Response 4:

We thank the reviewer’s suggestion. We have included the study's limitations in the paragraph before the conclusions section.

Comment 5:

In the reference list you have titles from 1974, please update the reference list as I am sure there are other newer articles on the subject.

Response 5:

We thank the reviewer’s suggestion. We have removed outdated references from the list.

Round 2

Reviewer 1 Report

Comments and Suggestions for Authors

I am agree with the publication 

Comments on the Quality of English Language

Minor editing of English language required.

Author Response

Comment 1:

Minor language editing is required.

Response 1:

We thank the reviewer’s suggestion. We have revised and refined the manuscript.

Reviewer 2 Report

Comments and Suggestions for Authors

The authors sufficiently address those concerns that I raised previously. Nice work. Just one small comment - the authors should consider to trim down the similarity score. As it stands now, 35% is unusually high for an original research article. 

Comments on the Quality of English Language

Minor language editing is required.

Author Response

Comment 1:

The authors sufficiently address those concerns that I raised previously. Nice work. Just one small comment - the authors should consider to trim down the similarity score. As it stands now, 35% is unusually high for an original research article.

Response 1:

We thank the reviewer’s suggestion. We have revised and refined the manuscript. We are very grateful for the requests and suggestions provided by the scholars. We have revised the manuscript accordingly to reduce similarity, especially in the sections highlighted in red.

Comment 2:

Minor language editing is required.

Response 2:

We thank the reviewer’s suggestion. We have revised and refined the manuscript.